# Analyzing Wav2Vec 1.0 Embeddings for Cross-Database Parkinson’s Disease Detection and Speech Features Extraction

**DOI:** 10.3390/s24175520

**Published:** 2024-08-26

**Authors:** Ondřej Klempíř, Radim Krupička

**Affiliations:** Department of Biomedical Informatics, Faculty of Biomedical Engineering, Czech Technical University in Prague, 16000 Prague, Czech Republic; klempond@fbmi.cvut.cz

**Keywords:** wav2vec, cross-database, classification, regression, feature importance, Parkinson’s disease

## Abstract

Advancements in deep learning speech representations have facilitated the effective use of extensive unlabeled speech datasets for Parkinson’s disease (PD) modeling with minimal annotated data. This study employs the non-fine-tuned wav2vec 1.0 architecture to develop machine learning models for PD speech diagnosis tasks, such as cross-database classification and regression to predict demographic and articulation characteristics. The primary aim is to analyze overlapping components within the embeddings on both classification and regression tasks, investigating whether latent speech representations in PD are shared across models, particularly for related tasks. Firstly, evaluation using three multi-language PD datasets showed that wav2vec accurately detected PD based on speech, outperforming feature extraction using mel-frequency cepstral coefficients in the proposed cross-database classification scenarios. In cross-database scenarios using Italian and English-read texts, wav2vec demonstrated performance comparable to intra-dataset evaluations. We also compared our cross-database findings against those of other related studies. Secondly, wav2vec proved effective in regression, modeling various quantitative speech characteristics related to articulation and aging. Ultimately, subsequent analysis of important features examined the presence of significant overlaps between classification and regression models. The feature importance experiments discovered shared features across trained models, with increased sharing for related tasks, further suggesting that wav2vec contributes to improved generalizability. The study proposes wav2vec embeddings as a next promising step toward a speech-based universal model to assist in the evaluation of PD.

## 1. Introduction

In recent years, the application of deep neural networks has revolutionized the field of medical research, offering innovative solutions to complex machine-learning challenges [1]. These networks have demonstrated state-of-the-art advancements across various medical domains [2,3]. The integration of deep learning (DL) models and wearable technology has emerged as a promising solution for the diagnosis and subsequent monitoring in medical contexts, particularly for Parkinson’s disease (PD) [4,5,6]. DL has proliferated in speech processing and language understanding. In addition to traditional methods based on calculating features from audio recordings [7], highly performing speech representations like wav2vec [8,9] have emerged, leading to improved performance in various tasks such as automatic speech recognition (ASR) [10] or emotion analysis from voice [11]. In the case of speech disorders, this includes specific applications for dysarthric speech classification [12]. An advanced application involves attempts to decode speech from brain activity [13]. While wav2vec has proven its capability to generalize across languages [14], an open question persists concerning PD, i.e., the determination of whether a particular subset of wav2vec features exhibits generalization across various diagnostic speech tasks.

PD stands as the second most prevalent degenerative disorder of the central nervous system, following Alzheimer’s disease. It affects about 1 in 100 people 65 years of age or older [15]. When someone under the age of 50 receives a diagnosis of PD, it is referred to as early-onset PD [16]. Its etiology is closely linked to the degeneration of dopamine-forming nerve cells within the substantia nigra, a critical component of the basal ganglia complex [17]. Speech, an integral indicator of motor functions and movement coordination, has proven to be exceptionally sensitive to central nervous system involvement [18]. Major movement symptoms of PD, including tremor, stiffness, slowing, and disturbances in posture and gait, only manifest when a substantial portion of brain cells are affected. In contrast, alterations in speech can occur significantly earlier, up to a decade prior to a formal diagnosis [19]. Recognizing speech impairment in the early stages of PD is crucial for a timely prognosis and the implementation of appropriate therapeutic measures [20]. The integration of advanced machine learning techniques with voice analysis, validated across three diverse datasets, has shown significant potential for enhancing early detection of PD [21].

This paper focuses on analyzing the feature importance of wav2vec 1.0 in PD speech and investigates whether the latent speech representations are shared across tasks, with increased sharing for related tasks. The study explores sets of features contributing across various datasets (participants rhythmically repeat syllable /pa/, Italian dataset and English dataset) and models, specifically in the context of healthy controls (HC) vs. PD classification, and regression tasks to predict demographic and articulation characteristics. Our study serves as a continuation of our prior work introduced in Klempir et al. [22], which analyzed wav2vec for classification. This paper incorporates additional cross-database, regression, and feature importance experiments.

The main contributions of this work are:Evaluation of wav2vec 1.0 embeddings for various inter-dataset HC vs. PD classification scenarios (also known as cross-db). There is still limited knowledge about high-performing methods for inter-dataset applications in PD detection based on speech. Research in this area is especially important for advancing federated learning.Current studies typically focus on diadochokinetic tasks or vowels. To address this gap, we demonstrate cross-db relevance using repeated syllables as well as read text.Providing evidence that wav2vec 1.0 embeddings can be used in various clinically relevant tasks related to PD, including regression. Our efforts focus on constructing models to predict quantitative targets such as age, loud region duration, and other articulation characteristics. Speech features like reading duration, pause detection, and speech instability are typically assessed through manual inspection and annotation of sound files using specialized software.To our knowledge, this paper is the first to investigate overlapping components in wav2vec 1.0 embeddings between classification and regression models, examining the presence of significant overlaps. Understanding the (dis)similarities between speech representations is essential for developing new corpus-independent models and for recommending standardized tasks that can be applied across different datasets.This work prioritizes wav2vec 1.0 over wav2vec 2.0, extensively examining its embeddings generated from its output layer. Despite its non-transformer-based design, wav2vec 1.0 has shown high performance, even comparable to other transformer-based embeddings.

The remainder of this paper is structured as follows: Section 2 reviews related work and further introduces the present problem. Section 3 details the methods used in this study, including the datasets and machine learning approaches applied. Section 4 presents the results and provides an analysis of the different experiments conducted. Section 5 offers a discussion of these results, and Section 6 concludes the paper with final remarks.

## 2. Related Work

The speech and voice characteristics of individuals with PD exhibit variations influenced by age and gender [23]. Early-onset PD, occurring before the age of 50, introduces an additional layer of complexity in the understanding of these nuances [16]. The precise prediction of both chronological and biological age in PD constitutes a clinically significant task. Recent machine learning studies have leveraged extensive datasets, such as the UK Biobank brain imaging data, to construct brain-age models and investigate various aging-related hypotheses [24,25]. One approach involves calculating the brain-age delta by subtracting chronological age from the estimated brain age [26]. In the context of PD, Eickhoff et al. conducted a comprehensive analysis on two PD cohorts (de novo and chronic), revealing a substantial increase in biological age—of approximately three years—compared to chronological age, a phenomenon evident even in de-novo patients. This age discrepancy significantly correlates with disease duration, as well as heightened cognitive and motor impairment [27]. The prediction of age based on speech features typically employs neural networks, demonstrating an efficient level of accuracy in classifying individuals into age groups. For instance, studies have reported testing accuracies as high as 85.7% [28], and classification errors by age group of less than 20% [29].

DL methods have exhibited their efficacy in extracting valuable features from voice and speech, particularly in the classification between those with neurological disorders and HC [30]. Convolutional neural networks (CNNs) stand out as a class of DL methods that surpass classical machine learning approaches in terms of performance. CNNs have achieved remarkable accuracy rates, often exceeding 99% [31,32]. Various CNN architectures are commonly employed in these applications, with data typically transformed into the time-frequency domain to retain both temporal and frequency information. This transformation can involve a spectrogram-based short-time Fourier transform (STFT), as observed in the analysis of PD vowel signals [32]. Additionally, CNNs can operate on mel-scale spectrograms of hyperkinetic dysphonia recordings [31] or utilize continuous wavelet transform to model articulation impairments in patients with PD [33]. Furthermore, research by Vásquez-Correa et al. demonstrated that fine-tuning CNNs through transfer learning contributes to enhanced accuracy in classifying patients with PD across different languages [34].

Recently, representation learning (RL) technology has facilitated the transformation of raw biomedical data into compact and low-dimensional vectors, commonly referred to as embeddings [35]. These embedding methods have gained widespread adoption across various biomedical domains, including natural language processing (NLP), where they are utilized to represent clinical concepts derived from unstructured clinical notes, such as symptoms and lab test results [36]. Compression algorithms, similar to DL autoencoders, have demonstrated efficacy in constructing embedding vectors for classification tasks, particularly in out-of-distribution domains [37]. Furthermore, NLP embedding techniques have been used to generate concise and scalable features for virome data, offering insights into the importance of each sequence position in the resulting supervised model outputs [38]. In RL for speech, embeddings serve as fixed-size acoustic representations for speech sequences of varying lengths [39]. The application of speech self-supervised learning (SSL) has enabled the utilization of large datasets containing unlabeled speech signals, achieving remarkable performance on speech-related tasks with minimal annotated data. These SSL embeddings systems have been comprehensively benchmarked, considering factors such as model parameters and accuracy across various tasks [40]. For accessibility, pre-trained ASR embeddings models are readily available through the Hugging Face repository [41].

Speech pre-trained embeddings relevant to the automatic assessment of PD include x-vectors [42], TRILLsson [43], HuBERT [44], and wav2vec [8,9]. In a comparative study by Favaro et al., the aforementioned architectures were evaluated for PD detection in multi-lingual scenarios, marking the initial application of TRILLsson and HuBERT in experiments related to PD speech recognition [45]. There are existing studies that show the capability of x-vectors for analyzing PD speech [46,47]. The method of wav2vec embeddings, which is available in two versions (wav2vec 1.0 [8] and the transformer-based wav2vec 2.0 [9]), represents a class of DL designed for self-supervised, high-performance speech processing. Fine-tuned wav2vec has proven efficient across various speech recognition tasks and languages [14]. The recent application of the transformer-based wav2vec 2.0 showcased its utility in developing speech-based age and gender prediction models, including cross-corpus evaluation, with significant improvements in recall compared to a classic modeling approach based on hand-crafted features [48]. Additionally, wav2vec 2.0 representations of speech were found to be more effective in distinguishing between PD and HC subjects compared to language representations, including word-embedding models [49]. As a pre-trained model, wav2vec shares the advantage with TRILLsson and x-vectors of being directly applicable without the need for further training, addressing the data-hungry nature common to many neural networks in the field.

Since 2023, the exploration of cross-database classification between HC and individuals with PD has attracted considerable attention [22,50]. Hires et al. conducted a study on the inter-dataset generalizability of both deep learning and shallow learning approaches, specifically focusing on sustained vowel phonation recordings. Their findings showed excellent performance during model validation on the same dataset, with a drop in accuracy during external validation [50]. Our previous study focused on the application of wav2vec 1.0 to reading speech, which presented a leave-one-group-out classification between Italian and English languages, with a relatively minor drop in performance observed [22]. The relevance of cross-database classification was emphasized by Javanmardi et al. [12], who investigated individual wav2vec layers for dysarthria detection. Their conclusions highlighted the need for further research to explore the generalizability of wav2vec features, particularly in cross-database scenarios. Furthermore, the study [50] found that distinct sets of features contribute differently across various datasets, and the same set of features is not universally shared across models for individual datasets. In 2024, Javanmardi et al. advanced the field of inter-dataset dysarthric speech detection by exploring fine-tuned wav2vec 2.0 across three dysarthria datasets with a layer-wise analysis [51]. The same research group also conducted a systematic comparison between conventional features and pre-trained embeddings for dysarthria detection, demonstrating that intra-dataset pre-trained embeddings outperformed conventional features [52].

The International Journal of Medical Informatics has introduced a comprehensive checklist for the (self)-assessment of medical artificial intelligence (AI) studies [53], emphasizing the utmost significance of feature importance and interpretability when introducing new methods for real-world applications. In the domain of speech recognition, mel-frequency cepstral coefficients (MFCCs) are emerging as key features for assessing speech impairments in neurological diseases; however, their interpretability remains limited [54]. First, experiments have been conducted to explain MFCCs and move closer to achieving interpretable MFCC speech biomarkers [55]. The study by Favaro et al. highlights another relevant finding: models based on non-interpretable features (DL embedding methods) outperformed interpretable ones [45]. Interpretable feature-based models offer valuable insights into speech and language deterioration, while non-interpretable feature-based models can achieve higher detection accuracy.

The remote assessment of (patho)physiological parameters, encompassing variables such as body temperature, blood oxygenation, heart rate, movement, and speech, holds significant relevance in scenarios where direct contact is impractical or must be avoided for various reasons. The collection of speech data, accessible to virtually anyone with an audio-enabled device, presents an opportunity to remotely screen for PD, thereby promoting inclusivity and accessibility in neurological care [23]. In contemporary biomedical AI systems, particularly multimodal ones, an abundance of distinctive features or signals is embedded within the dataset [56]. Speech and voice play an important role in numerous instances [57]. To get the full potential of these complex datasets, advanced AI techniques can be employed to align multimodal features onto a shared latent space. This approach enhances the precision of phenotype prediction for highly heterogeneous data spanning various modalities [58]. We anticipate that the mentioned AI systems can be improved by incorporating speech representations like wav2vec. However, it is essential to conduct an investigation into the wav2vec component, examining its capabilities to generalize across different datasets and tasks.

## 3. Methods

### 3.1. Datasets

The study utilized recordings and metadata from three distinct datasets for classification and regression experiments: a rhythmic syllable repetition dataset (used for classification) and both Italian and English datasets (used for classification and regression in the case of Italian and classification for English).

#### 3.1.1. Participants Rhythmically Repeat Syllables /pa/

This dataset involved 30 male PD patients and 30 male age-matched HC Czech participants rhythmically repeating the syllable /pa/ [7]. The data, consisting of audio signals with a sampling frequency of 48 kHz, focused primarily on evaluating “pa” recordings, a standardized speech examination in PD. This dataset was used for training classification models, with a total sample size of 60 subjects (30 HC vs. 30 PD).

#### 3.1.2. Italian Study by Dimauro et al. [59]

This dataset, used for both classification and regression tasks, stems from a study assessing speech intelligibility in PD conducted at the Università degli Studi di Bari, Italy [59]. The dataset used for HC vs. PD classification included 50 subjects (elderly HC: n = 22, PD: n = 28), with measurements of text readings (44.1 kHz) available for each individual. For the regression experiments, we also included a young HC group (young HC: n = 15), along with metadata for regression tasks such as age and the estimated number of characters read per second (age, characters per second). In the HC group, individuals aged 60–77 years were included, consisting of 10 men and 12 women. None of these individuals reported any particular speech or language disorders. Regarding the PD group, patients aged 40–80 years were included, consisting of 19 men and 9 women. In the young group, individuals aged 19–29 years were included, consisting of 13 men and 2 women.

In total, this dataset included 50 subjects for classification (22 HC vs. 28 PD). For regression tasks, which involved modeling age and articulation characteristics, the inclusion of young, healthy controls increased the total number of subjects to 65.

#### 3.1.3. English Dataset

Chosen for classification experiments, the English dataset “Mobile Device Voice Recordings at King’s College London (MDVR-KCL)” [60] comprised 21 HC and 16 PD subjects who read aloud “The North Wind and the Sun”. Measurements of text readings were available for each participant, with a sampling frequency of 44.1 kHz. There is no available information regarding the distribution of gender and age. This dataset was used for training classification models, with a total sample size of 37 subjects (21 HC vs. 16 PD).

### 3.2. Signal Processing and Feature Extraction

#### 3.2.1. Naive Loud Regions Segmentation

We derived an additional parameter for the purposes of regression tasks, which we refer to as loud region duration. Loud region duration is a parameter that can be approximated from raw signals and is directly associated with reading duration. Speech characteristics such as reading duration, pause determination, and speech instability are important factors in PD and are typically correlated with clinical scales as markers of motor impairment [61]. These parameters are typically calculated through manual inspection and annotation of sound files using the Praat software (current release, version 6.4.18).

The preprocessing phase involved segmenting the signals into binary loud regions. As a basic baseline approach, the raw audio signal was segmented using a straightforward formula, creating a binary vector representing quiet and loud regions. Each data point in the series was encoded as 1 if its absolute value exceeded the mean absolute value of the signal and 0 otherwise. Subsequently, the binary vector underwent averaging through a 10,000-sample window rolling average and was summed to generate a single numerical value for each recording.

#### 3.2.2. MFCCs Features Calculation

To establish baseline comparisons, we calculated the audio features for the recordings stored in WAV files using the Librosa Python library [62]. The recordings were resampled at 16 kHz. Subsequently, the waveform underwent processing with Librosa (version 0.10.2) to compute 50 MFCCs, with each coefficient averaged over time (MFCC-mean). This computation resulted in a vector comprising 50 values.

#### 3.2.3. Wav2Vec Embedding and Features Calculation

To assess the generalization capability across databases and minimize the need for feature engineering, we employed wav2vec 1.0 [8], a speech model that accepts a float array corresponding to the raw waveform of the speech signal. All recordings were resampled at 16 kHz, a prerequisite for the wav2vec embedding used for feature extraction. In contrast to traditional signal processing methods, wav2vec is capable of learning a suitable representation of an audio signal directly from data, avoiding the necessity of manual feature extraction [22].

For our experiments, we utilized pretrained components of the wav2vec model, which maps raw audio to a latent feature representation. This model is pretrained in an unsupervised manner and has demonstrated advanced performance in speech recognition tasks with minimal fine-tuning. Specifically, we obtained a publicly available wav2vec-large model trained with the LibriSpeech training corpus, which contained 960 h of 16 kHz English speech [63].

Due to the dynamic representation of the wav2vec embedding over time [22], we computed a derived 1D static feature vector of 512 dimensions for each recording based on three different statistics (mean: wav2vec-mean, standard deviation: wav2vec-std, sum: wav2vec-sum) along the time axis. The output of this method served as the input for machine learning models for classification and regression. Similar approaches exist in the literature to aggregate specific DL audio embeddings, as seen in TRILLsson and wav2vec 2.0 [45]. Since feature extraction from long recordings requires relatively complex computation in [45], the authors divided long recordings into 10-s segments. However, with wav2vec for this research, we found this segmentation to be unnecessary, allowing us to process the entire signal without the need to split it into sections.

In an extension of our previous research [22], this study enriches the previous results by incorporating wav2vec-std. Additionally, a derived feature vector of length 20 is calculated, incorporating 10 principal component analysis (PCA) components of wav2vec context (wav2vec-mean) and 10 PCA components of spectral MFCC characteristics (MFCC-mean) combined into a single vector. Ten components were selected to capture a minimum of 90% of the original variance (cumulative sum of explained variance) across datasets for both wav2vec-mean and MFCC-mean. We aim for a compact resulting vector length with an equal number of components for both input vectors.

### 3.3. Modeling and Statistical Methods

A high-level overview of the proposed machine learning methodology is illustrated in Figure 1.

#### 3.3.1. Cross-Database Classification Experiments

To assess the generalizability of wav2vec and MFCCs features across multiple PD detection datasets, we conducted a series of classification experiments (categorizing HC as negative and all PD cases as positive). This experimental setup is entirely unique in this version of the paper compared to our prior publication [22]. Following a similar approach as recently presented in [50], we adopted combo scenarios. Specifically, we assessed performance on individual datasets, and more importantly, we implemented additional scenarios: one that utilizes a combination of multiple datasets for training and evaluates a completely unseen remaining dataset, and vice versa. For binary classification modeling, we employed an ensemble random forest classifier (with default settings, n_estimators = 100 and criterion = ‘gini’) from the Python scikit-learn library [64,65]. For selected cross-database results, we implemented a binary classifier using XGBoost, configured with the parameters “softmax” and eval_metric = [‘merror’, ‘mlogloss’] [66]. Model evaluation was conducted using 5-fold cross-validation with 5 repeated fits to measure the performance of the classification models. The interpretation of the classification results involved comparing the model output with the ground truth using a receiver operating characteristic (ROC), where the area under the curve (AUC) was employed to quantify the level of accuracy. In this study, the primary evaluation metric for model performance was the area under the ROC curve (AUROC).

#### 3.3.2. Regression Experiments

We utilized wav2vec features in regression modeling tasks for modeling age and parameters associated with articulation rate. In the modeling phase, we employed the lasso with alpha = 0.01 (least absolute shrinkage and selection operator) model from the Python scikit-learn library [64,67] and conducted model evaluations using 5-fold cross-validation. Lasso employs regularization through a geometric sequence of Lambda values. Its primary advantage lies in the suppression of redundant features, contributing to improved model generalization. This characteristic proved particularly beneficial when dealing with a limited number of observations and a substantial number of features. To measure the performance of our model, the following metrics were computed: Spearman correlation coefficient (rho) [68], r-squared (r2) metric [69], and mean absolute error (MAE) [70].

#### 3.3.3. Strategy to Evaluate Overlapping Components across Models

To explore the hypothesis of a potential link between classification and regression models, we investigated the feature importance of the wav2vec embedding as an important aspect of interpretability. While wav2vec exhibits promising potential for applications in speech processing for neurological diseases, it is essential to grasp its behavior across various models and tasks. The goal was to identify features that exhibit common features across models and evaluate their (dis)similarity. Approaches that involve models designed to handle irrelevant and unrelated features of the modeled variable can lead to enhanced model explainability. Such examples are linear regression, where coefficients can be set to zero, or decision trees, which do not use irrelevant features in tree splitting criteria. The model choice not only improves model interpretability but also reduces computational requirements for training and model utilization. Our objective was to analyze global feature importance, which refers to the overall significance of a feature across all instances within the dataset. It is essential to note that there is no universally superior method for determining global feature importance, as each method provides estimates based on distinct assumptions.

In the initial step, we identified the top 30 most contributing wav2vec features for each model considered. For classification, logistic regression, known for its feature importance computation built-in functionalities [71], was employed. For regression, we utilized the random forest regressor, followed by the SHAP explainability method [72]. SHAP values were extracted from the best-fitting model for all individuals. In cases of more complex models, Shapley values are approximated through Monte Carlo sampling. The top features were obtained based on the importance of each factor, ranked from most to least important according to feature importance values. The calculation of feature importances for both built-in scikit-learn tools and SHAP typically involves utilizing the training set.

In the subsequent step, the resulting top 30 most important features for each model were then subjected to statistical testing to assess the significant coverage of common features shared between models. The Fisher exact test was used to compute *p*-values. The computation considered the number of (non)-overlapping features and the total number of features. P-values below 0.05 were considered statistically significant. Visualization using Venn diagrams was used to illustrate the unique/common sets of features observed among the models. As an alternative approach to evaluating significance, we conducted a simulation experiment to identify the minimum number of shared features considered significant. In 10,000 runs, two sets of 30 features were randomly generated from a pool of 512 total features, and their intersection was calculated. The mean number of shared features was 1.76, with 5 shared features (99th percentile) demonstrating statistical significance (associated with a *p*-value of 0.01). To address potential concerns related to multiple comparisons, we applied the Bonferroni correction to establish the corrected threshold for significance. A significance threshold was maintained, with a minimum of 6 shared features remaining statistically significant after correction. This corresponded to a *p*-value of 0.013, obtained by multiplying the base *p*-value of 0.004 by a factor of 3 (reflecting the number of independent tests for each presented Venn diagram).

## 4. Results

The overall distributions of the recording durations for the three datasets are visualized in Figure 2. All three datasets were utilized for classification tasks (i.e., HC vs. PD). Additionally, the Italian dataset was employed for regression experiments. Young Healthy Controls (yHC) from the Italian dataset were excluded from the binary classification experiments due to their significantly different age distribution.

### 4.1. Intra- and Inter-Dataset Classification for Detection of PD

This section presents the outcomes of the four primary classification scenarios detailed in Table 1. The table reports the AUROC scores achieved by the respective models on individual datasets (intra-dataset classification) and on combined datasets with external evaluation (inter-dataset classification). A subset of the results discussed here was previously addressed in [22], and these results are appropriately marked in Table 1.

The first scenario involved models trained on individual datasets and evaluated on the same datasets, covering three cases—/pa/, Italian, and English datasets separately. For /pa/, the top-performing model was achieved with 10 PCA MFCC-mean, slightly outperforming the wav2vec-sum. The combination of wav2vec and MFCC features did not enhance overall performance. Remarkably, an excellent performance was observed across all models for the Italian dataset. In the case of the English dataset, AUROCs varied, with 0.74 for the MFCC-mean model and the best-performing model reaching 0.80 for wav2vec-std.

The second scenario examined the performance of the three models on combined datasets, where two datasets were mixed to create a larger dataset. The aim was to increase the volume of training data, potentially enhancing model performance. When combining the Italian and English datasets, overall, wav2vec yielded superior results compared to MFCCs. It is noteworthy that the mixing of wav2vec and MFCC PCA components elevated the AUROC to 0.88, surpassing the individual AUROC values of 0.83 for wav2vec-mean and 0.81 for MFCC-mean. Conversely, in the case of merging Italian and /pa/ datasets, MFCCs outperformed wav2vec parameters. Additionally, for the mixed case of English and /pa/ datasets, the 10 PCA MFCC-mean exhibited a higher AUROC (by 5–10%) compared to other models.

Scenario 3 comprised the most extensive set of results. A total of six models were trained to assess cross-database generalizability. When trained on /pa/, wav2vec features failed to accurately classify HC vs. PD in both the Italian and English datasets. Some non-random generalizability signals (AUROC = 0.68) were observed in the cases of MFCC-mean and 10 PCA wav2vec-mean in the Italian dataset. The same AUROC was achieved for the English dataset in the case of 10 PCA MFCC-mean. When trained on Italian data, the combination of PCA components of wav2vec and MFCC features proved to be the most effective when evaluating /pa/. Evaluating the Italian model on the English dataset revealed that wav2vec, previously shown to be promising in detecting HC vs. PD, outperformed the newly introduced MFCC-based features. A similar trend was observed in the reverse case, i.e., training on English and evaluating on Italian. Lastly, evaluating the English model on /pa/ resulted in decreased accuracy.

In Scenario 4, all wav2vec models trained on mixed English and /pa/ datasets achieved an AUROC higher than 0.8. The minimum was 0.8 for wav2vec-sum, and the maximum was 0.88 for wav2vec-std. Otherwise, we noticed analogous trends as those observed in Scenario 3.

The selected cross-database results obtained on individual datasets were highly satisfactory, indicating a non-random level of generalizability in language-independent models for discriminating between PD and HC (Figure 3, Figure 4 and Figure 5).

### 4.2. Regression Models to Predict Age and Articulation

The next section provides an examination of the results obtained for patient characteristics from the Italian dataset (yHC, HC, PD; N = 65). Initially, we explored paired correlations among age, characters per second, and duration parameters with naive loud region segmentation (Figure 6A). Subsequently, we utilized wav2vec-mean, and the features were reduced to two dimensions using PCA. The PCA visualizations (Figure 6B) of the wav2vec-mean embedding illustrated the correlation structure with the parameters from Figure 6A. Individual PCA plots, color coded with the respective parameter values, revealed patterns where similar values clustered closer to each other, confirming that capturing the age and articulation parameters based on wav2vec is highly likely. Furthermore, we visually examined the impact of gender on age prediction by previewing 2 PCA components of the wav2vec-mean (Figure 7).

The trained wav2vec-mean regression models demonstrated correlations with age (Spearman R = 0.56), characters per second (Spearman R = 0.74), and loud region duration (Spearman R = 0.84). Additional metrics and the reported performance of the trained lasso regression models are presented in Figure 8.

### 4.3. Exploration of Overlapping Important Features across Models

Feature importance assessments were conducted for various classification and regression models utilizing wav2vec-mean features. We focused on the top 30 impactful features, ranked from the most influential to the least, out of the complete set of 512 features. The connections between classification and regression model profiles are presented in Figure 9A and Figure 9B, respectively. Overall, the Fisher exact tests indicated a significant number of common features in all model pairs (*p* < 0.05). In the binary classification models (Figure 9A), the strongest association was identified between the Italian and English speech datasets (11 shared features). For regression (Figure 9B), the overlap between the model explaining loud region duration and the model explaining characters per second was more pronounced (10 shared features) than with the model explaining age (5 shared features). The increased number of shared features between loud region duration and characters per second aligns with the observed correlation coefficients between loud region duration vs. characters per second (Spearman R = −0.76), compared to the correlation between loud region duration vs. age (Spearman R = 0.53), as shown in Figure 6A. Taking into account the stringent Bonferroni corrected comparisons, certain pairs that had shown significance using the Fisher exact test were no longer significant (i.e., had fewer than 6 shared features). This applied specifically to the classification comparison between /pa/ and English, as well as both articulation rate models versus age.

Furthermore, we investigated the potential overlap between classification and regression tasks in the context of a binary age classification model versus three regression tasks (Figure 10). The parameter age was considered in two distinct forms: firstly, as a binary class (young vs. elderly), and secondly, as a quantitative regression model predicting age. Concurrently, two additional models, addressing entirely different aspects, were examined and tested against the important features derived from the age classification model. The association between the number of features in age:classification vs. age:regression model was found to be significant, indicating shared features between the tasks. In contrast, other cases were not found to be statistically significant or over-represented.

To compare wav2vec-mean features with MFCC and to explore the effect of using a different classifier (random forest classifier) instead of logistic regression, we conducted two additional experiments. These experiments were performed for both wav2vec-mean (Table 2) and MFCC-mean (Table 3) features.

## 5. Discussion

This paper introduces a study on using wav2vec 1.0 embeddings and MFCCs to detect Parkinson’s disease from speech, focusing on cross-corpora classification and regression tasks. It demonstrates wav2vec’s performance in PD detection over traditional methods and explores its generalizability across languages and tasks.

In addressing the primary hypothesis concerning the exploration of shared common features, our study focuses on the feasibility of achieving satisfactory results through cross-database classification. The main component is covered by machine learning, where we aimed to discern patterns and features that exhibited consistency across diverse databases. We constructed multiple models through various train–test combinations to evaluate the efficacy of wav2vec as a robust speech representation for PD detection. In the combined training approach, we did not account for the heterogeneity in recording conditions, speech tasks, and participant demographics across the datasets—specifically, the rhythmic repetition of syllables in the first dataset, text readings in the Italian and English datasets, and variations in age and gender distribution. We consider this a possible example of corpus-dependent clustering, as defined in [73].

The main classification findings in Table 1 revealed that external validation did not result in a significant decrease in the AUROC metric for specific scenarios, particularly, when training on English and evaluating on Italian (AUROC > 0.90). This marked improvement stands in contrast to the baseline MFCC features (AUROC = 0.51). Additionally, other new findings were observed. Due to the high correlation among the wav2vec embedding features, we performed a calculation of 10 PCA. This approach demonstrated superior results in some cases, outperforming models trained on the full list of features, especially in conjunction with baseline MFCCs and /pa/ dataset. We observed that wav2vec-mean excelled in the classification of read text, whereas a wav2vec-MFCC combination proved partially effective for tasks involving repeating syllables (/pa/). This behavior suggests that wav2vec is well suited for external validation of complex speech tasks, while the combination of wav2vec and MFCCs performs better in simpler voice tasks. In conclusion, our classification experiments underscore the utility of wav2vec in cross-database PD detection.

In connection with existing peer-reviewed publications, our findings share relevance with the study conducted by Hires et al. [50]. The paper investigated the inter-dataset generalization of traditional and CNN machine learning approaches for PD detection based on voice recordings, specifically focusing on vowels. In their study, external validation demonstrated a decline in accuracy, with the highest AUC surpassing 0.7 in certain instances. Our research adopts a similar validation strategy, but we did not perform any hyperparameter tuning. In a recent study [45], the assessment of PD using wav2vec 2.0 and other DL embeddings in a cross-lingual context was explored. The investigation compared interpretable and non-interpretable features. Relevant to our results, the study’s cross-lingual experiments revealed significant performance variations based on the target language, emphasizing that DL embeddings can achieve an AUC exceeding 0.9 in cross-language validation. This finding supports our results obtained from wav2vec 1.0 and suggests possible alignment with other DL embeddings, such as TRILLsson. Although comparing results accurately is challenging due to differences within the datasets, we evaluated our best-obtained results against previous works that developed HC vs. dysarthria models for inter-dataset (i.e., cross-database) scenarios. The findings from this comparison are summarized in Table 4.

In a few scenarios presented in Hires et al. [50], they utilize the same Italian dataset [59] as in this study, along with the Czech Parkinsonian dataset (not the one used here), for both intra- and inter-dataset classification. The results are comparable and consistent with our findings in terms of demonstrating very high accuracy for intra-dataset discrimination. Regarding other similar studies that have addressed classification using the same datasets (only intra-dataset), for the Italian dataset, we refer to the work by Malekroodi et al. [75]. The authors published a table comparing the accuracy results obtained from six approaches across five studies, all of which achieved an accuracy exceeding 95%. For the English dataset, we reference Cesare et al. [76], who compared multiple signal representations and classifiers, consistently obtaining accuracies close to or exceeding 90% in intra-dataset scenarios.

The analysis of overlapping components within the embeddings across datasets revealed that there is not a universally identical set of features shared across models for individual datasets. This variability is expected, especially when dealing with hundreds of features. Given the substantial number of features within wav2vec (the same applies to other DL embeddings as well), our focus was on the most impactful ones, extracting the top 30 features for each model. Ranking the top 30 or top 50 features (e.g., out of hundreds of features in total) based on feature importance, followed by the use of statistical methods to evaluate overlaps or differences between conditions, is a well-established approach in biomedical research studies [77,78,79]. A strong cross-language signal pattern was observed between Italian and English speech classification models, indicating an enhanced level of generalizability across languages, specifically in the classification of PD versus healthy control—a trend consistent with our observed cross-database classification results. This finding aligns with existing research [80], demonstrating that unsupervised pretraining effectively transfers across languages.

Emphasizing the variability of embedding features depending on the training data, the additional experiments involved exploring the number of overlapping sets of features across 30 runs, each with varying numbers of top features. The analysis indicated that a portion of overlaps initially identified using logistic regression lost their significance when re-evaluated with the random forest classifier. Nevertheless, the wav2vec-mean features continued to detect significant overlaps between the Italian and English models. Taken together, there was a greater overlap between sets of features when comparing wav2vec with MFCC, with wav2vec showing more prominent overlaps. Furthermore, a higher number of overlaps was observed in regression tasks where models were created within a single dataset.

Identifying the most important features in DL embeddings holds the potential to interpret DL model outcomes by exploring cross-task relationships, enhance model performance, streamline model complexity by removing less critical features, mitigate overfitting, and further improve the generalization of PD detection from speech. However, given that the embedding as a whole constitutes the feature, interpreting individual neurons within an embedding by means of feature importances can be challenging. The results presented here should be considered preliminary, and we recommend further research to conduct a more detailed disentangled analysis to better understand the significance and interpretability of these embeddings. In addition, in the context of our research, where we utilized MFCC analysis as a baseline, Rahman et al. conducted a feature importance analysis on MFCCs [23]. Their investigation revealed that features that had an impact on the model’s performance were typically spectral features.

The findings of this study hold potential implications for broader applications in diverse contexts. As DL in PD continues to advance, overcoming technical and methodological challenges is crucial for widespread technology adoption. Sigcha et al. have highlighted the importance of large unsupervised data collection and the use of semi-supervised deep learning approaches to enhance PD symptom detection and severity estimation, ultimately improving the generalization capability of developed solutions [4]. The deployment of the wav2vec in a multimodal system, either separately as a single component or in combination with MFCCs, offers opportunities for external validation and performance enhancement. For instance, incorporating Mel spectrograms for the audio component, as demonstrated in [57], can contribute to the development of multimodal biomedical AI. Additionally, the results obtained in remote speech PD, combined with a remote system like finger tapping [81], open possibilities for diverse applications at home. Moreover, utilizing federated learning for PD detection, locally extracted using the wav2vec 2.0 model, encourages further exploration of techniques for the generalization of models from databases of the same pathology, in different languages, without the need for sharing information between other institutions [82].

This paper has some limitations. First, our exploration solely focused on one wav2vec architecture, disregarding newer embeddings like wav2vec 2.0, which has demonstrated a slight improvement (2–3% in average recall) over wav2vec 1.0 in emotion recognition [11]. Although the difference between wav2vec versions is not substantial, there is potential for enhancement. Secondly, it is important to note that caution should be exercised in interpreting wav2vec representations in specific cases, such as slow or fast speakers, where model predictions may be less accurate. Further investigation and error analysis, involving additional datasets, are warranted for confirmation. Finally, the regression model for the age parameter in the Italian dataset reveals an apparent gap between young and elderly individuals, leaving some age ranges uncovered. This gap could introduce bias in the accuracy of the trained model, potentially leading to failures in predicting unseen data. Retraining the model separately for young and elderly groups might be necessary to address this limitation.

For future work, we can consider the implications of training the wav2vec model exclusively on English data and whether experiments with a multilingual version could offer additional insights. Extending our experiments to include testing generalizability in other out-of-domain datasets, such as snoring [83], would be beneficial. Additionally, we anticipate further enhancements by incorporating proper augmentation techniques, such as audio-specific methods like noise addition or volume control, to introduce variability to the recordings and perturb the models. Initial attempts related to wav2vec augmentation have been explored in [84], and a recent study has provided a comparison of data augmentation methods in voice pathology detection [85]. Furthermore, the role of augmentation in wav2vec model training itself (wav2vec-aug) has demonstrated a substantial improvement in accuracy [86].

## 6. Conclusions

Based on the experimental results, this study showed the suitability of wav2vec for accurate intra-dataset classification and satisfactory performance in inter-dataset classification. It has proven effective in diverse regression tasks, such as modeling age and articulation characteristics like characters per second or loud region duration. Of utmost significance, the analysis of global feature importances reveals shared features among classification datasets, regression models, and statistically significant links between classification and regression tasks. While it cannot be definitively concluded that only a small subset of the 512 wav2vec features would universally solve any classification or regression task, the observed pattern suggests that similar tasks share common features. Transferability was more pronounced for similar speech tasks, such as reading or spoken text, as evident in both shared feature importances and cross-database classification. The concluding insights emphasize the importance of feature importance methods in interpreting the generalization capability of developed deep learning solutions. The study proposes wav2vec embeddings as a next promising step toward a speech-based universal model to assist in PD evaluation.

In our future research, we aim to address some of the limitations reported in this study. Specifically, we plan to validate our findings on more extensive datasets and further explore strategies for analyzing overlapping components between models and tasks. We will continue evaluating pretrained embeddings (such as wav2vec 1.0, wav2vec 2.0, and HuBERT) as universal representations that can accurately solve various problems in PD. Additionally, we intend to conduct a systematic comparison of different classification and regression techniques, followed by hyperparameter optimization.

## Figures and Tables

**Figure 1 sensors-24-05520-f001:**
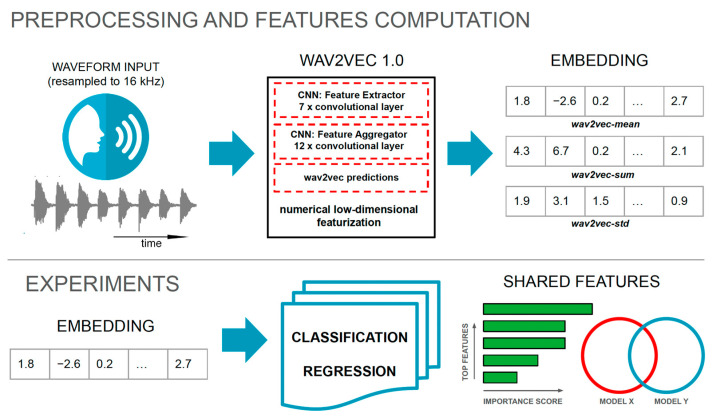
Illustrative diagram of the proposed signal processing and experiments.

**Figure 2 sensors-24-05520-f002:**
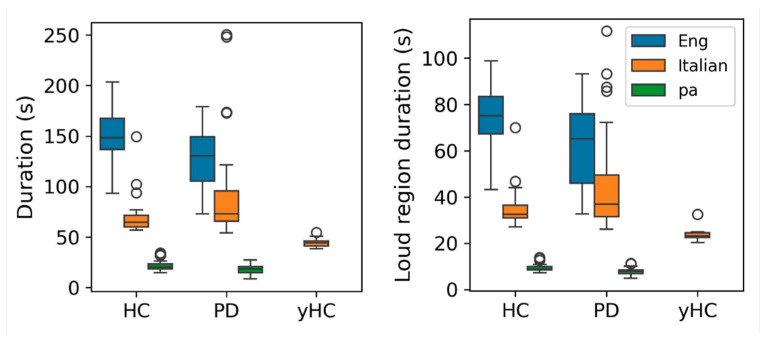
Distributions of the recording durations (**left**) and loud region durations (**right**) across the datasets used in the study.

**Figure 3 sensors-24-05520-f003:**
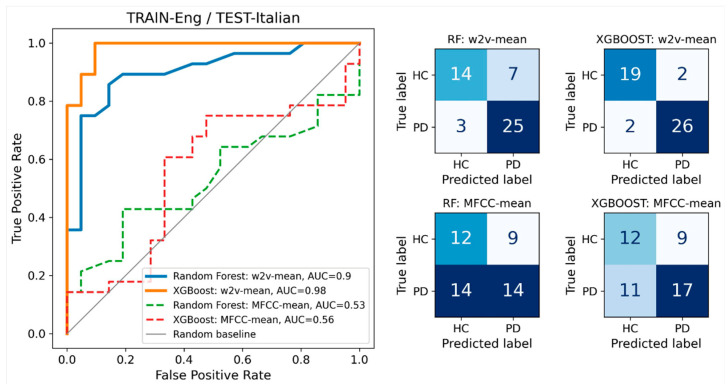
TRAIN-Eng/TEST-Italian: AUROC curves for the two speech representations and two classifiers. The plot on the left shows that wav2vec outperformed MFCC when trained on English data and tested on Italian data. XGBoost further improved the performance observed with random forest. Detailed classification results for each classifier are presented in the corresponding confusion matrix on the right. In this plot, the results were not cross-validated; instead, the classifier was trained on the entire dataset #1 and tested on the entire dataset #2.

**Figure 4 sensors-24-05520-f004:**
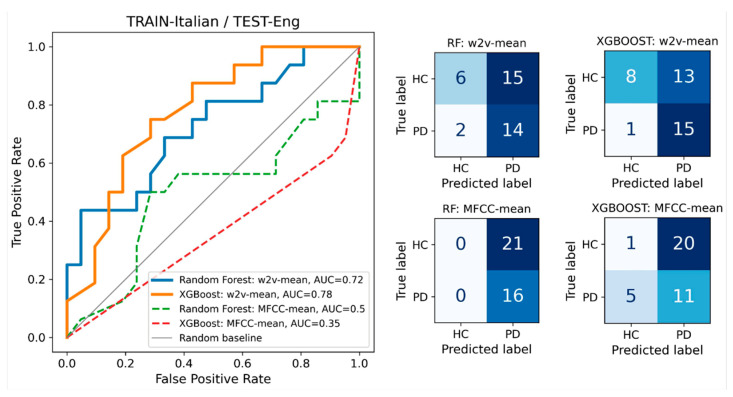
TRAIN-Italian/TEST-Eng: AUROC curves for the two speech representations and two classifiers. The plot on the left shows that wav2vec outperformed MFCC when trained on Italian data and tested on English data. Similar to the observations in Figure 3, XGBoost demonstrated improved performance compared to the random forest classifier. Detailed classification results for each classifier are presented in the corresponding confusion matrix on the right. In this plot, the results were not cross-validated; instead, the classifier was trained on the entire dataset #1 and tested on the entire dataset #2.

**Figure 5 sensors-24-05520-f005:**
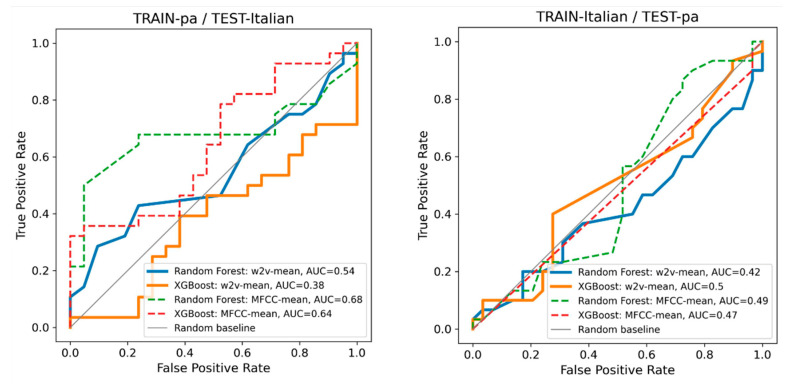
AUROC curves for the /pa/ dataset and the read text from the Italian dataset. The left plot shows that wav2vec failed in this prediction task when trained on /pa/ data and tested on the Italian data, and vice versa, as shown on the right. On the left, MFCC performed slightly better compared to wav2vec; /pa/ dataset is not very similar to the other two datasets, and it contributed little to the cross-db classification. The results were not cross-validated; the classifiers were trained on the entire dataset #1 and tested on the entire dataset #2.

**Figure 6 sensors-24-05520-f006:**
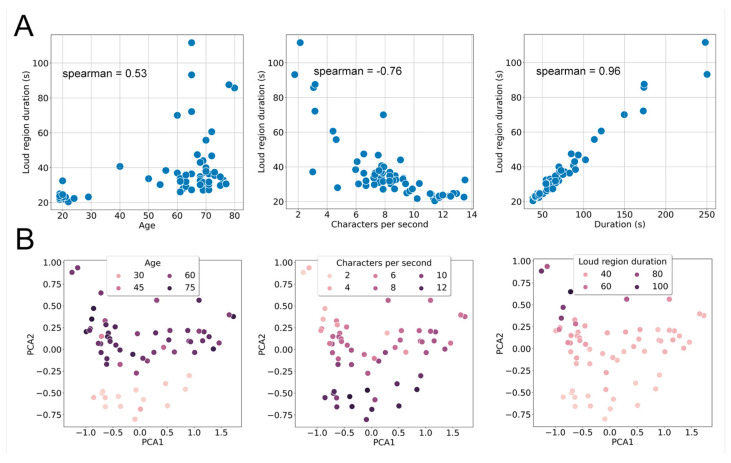
(**A**) Correlations between the features from the Italian dataset and the calculated duration of loud regions. (**B**) PCA plots depicting the relationship between wav2vec-mean features. The close grouping of similar colors suggests a nonrandom signal across all wav2vec-mean features. Each data point corresponds to one subject.

**Figure 7 sensors-24-05520-f007:**
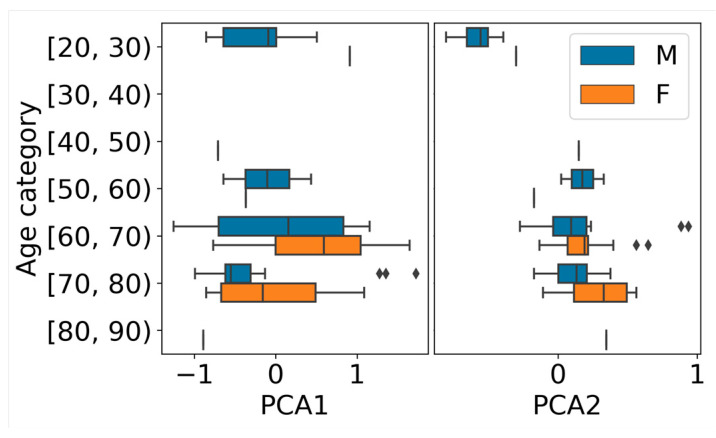
Boxplots illustrating 2 PCA components of wav2vec-mean, organized into distinct classes based on age and presented in separate panels for each component. The plots emphasize gender-related differences.

**Figure 8 sensors-24-05520-f008:**
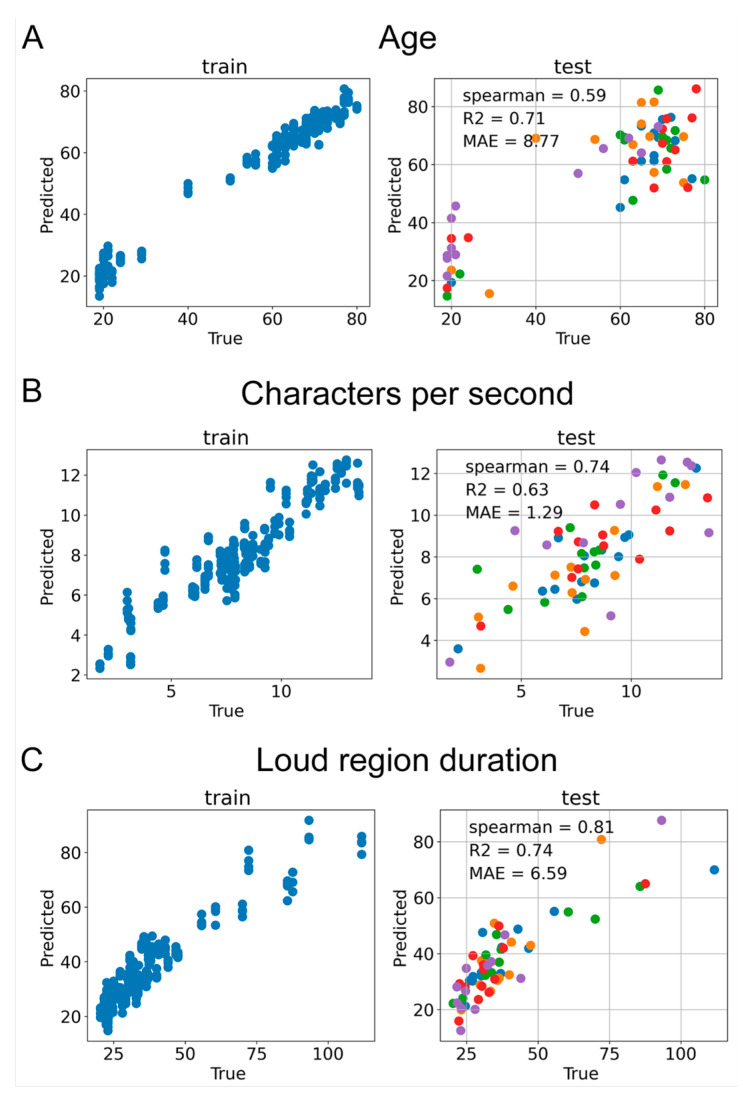
Performances of audio-wav2vec-based regression models on both training and testing data. Each distinct color in the plot corresponds to an individual run of cross-validation, specifically: (**A**) modeling age, (**B**) modeling characters per second, and (**C**) modeling loud region duration.

**Figure 9 sensors-24-05520-f009:**
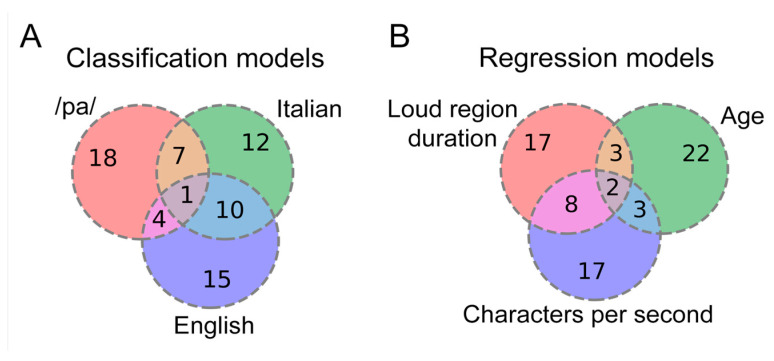
Venn diagrams illustrating shared features among multiple models: (**A**) classification and (**B**) regression.

**Figure 10 sensors-24-05520-f010:**
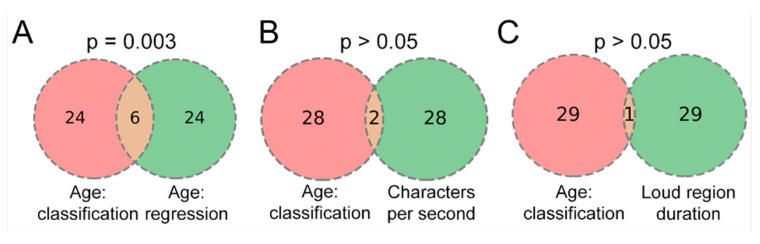
Comparative analysis of important features identified by the binary age classification model vs. top features derived from three regression models: (**A**) age:classification vs. age:regression, (**B**) age:classification vs. characters per second:regression, (**C**) age:classification vs. loud region duration:regression.

**Table 1 sensors-24-05520-t001:** The AUROC scores represent the classification performance of the respective models in distinguishing between HC and individuals with PD across four scenarios, including intra-dataset and inter-dataset comparisons; wav2vec is abbreviated to w2v.

#	TRAIN	TEST	w2v-Mean	w2v-Sum	w2v-std	MFCC-Mean	10 PCA w2v-Mean	10 PCA MFCC-Mean	MFCC-Mean + w2v-Mean **
1	pa	pa	0.61 *	0.82 *	0.70	0.78	0.58	0.84	0.81
	Italian	Italian	0.98 *	0.97 *	0.98	0.98	0.98	0.99	0.99
	Eng	Eng	0.80	0.71	0.80	0.74	0.65	0.79	0.77
2	Italian, Eng	Italian, Eng	0.94	0.90	0.93	0.89	0.83	0.81	0.88
	Italian, pa	Italian, pa	0.87	0.89	0.86	0.92	0.81	0.89	0.91
	Eng, pa	Eng, pa	0.72	0.75	0.71	0.71	0.58	0.81	0.78
3	pa	Italian	0.46	0.50	0.50	0.68	0.68	0.40	0.57
	pa	Eng	0.40	0.50	0.51	0.49	0.34	0.68	0.55
	Italian	pa	0.48	0.43	0.46	0.49	0.61	0.63	0.68
	Italian	Eng	0.72 *	0.56	0.67	0.47	0.56	0.42	0.51
	Eng	Italian	0.90 *	0.78	0.77	0.51	0.73	0.34	0.47
	Eng	pa	0.46	0.53	0.48	0.44	0.48	0.60	0.60
4	Italian, Eng	pa	0.49	0.43	0.52	0.55	0.53	0.63	0.64
	Italian, pa	Eng	0.72	0.61	0.61	0.56	0.46	0.46	0.51
	Eng, pa	Italian	0.86	0.80	0.88	0.56	0.75	0.50	0.64

* Results presented and discussed in [22]. ** 10 PCA from w2v-mean + 10 PCA from MFCC-mean.

**Table 2 sensors-24-05520-t002:** Analysis of overlapping wav2vec-mean components, experiment using 30 repeats. The values in the table represent the number of features that overlapped within the model under the given scenario (presented as mean (std)). The column values indicate the number of top features with overlap determined as the minimum threshold for significance. The maximum number of features for wav2vec was 512. The absolute values of the feature importances were not zero. Abbreviations used: CHPS: characters per second, LRD: loud region duration.

Task	Scenario	TOP10 (min. 2)	TOP20 (min. 3)	TOP30 (min. 5)	TOP50 (min. 9)
Random Forest Classifier	pa vs. Italian	0.1 (±0.3)	0.4 (±0.7)	1.2 (±1.0)	3.9 (±1.8)
	pa vs. Eng	0.0 (±0.2)	0.4 (±0.6)	1.0 (±0.9)	4.5 (±1.8)
	Italian vs. Eng	0.8 (±0.8)	1.8 (±1.0)	* 3.9 (±1.2)	* 7.6 (±1.8)
Random Forest Regressor	Age vs. CHPS	** 2.1 (±0.7)	** 4.7 (±1.4)	** 7.6 (±1.5)	** 11.4 (±2.0)
	Age vs. LRD	0.5 (±0.5)	* 2.1 (±1.0)	* 4.5 (±1.4)	* 7.1 (±2.5)
	CHPS vs. LRD	0.8 (±0.7)	** 3.8 (±1.3)	** 8.5 (±1.7)	** 16.4 (±2.2)

* Indicates that significant overlap was achieved at least once by the threshold mean + 1σ. ** Indicates that significant overlap was achieved on average.

**Table 3 sensors-24-05520-t003:** Analysis of overlapping MFCC-mean components, experiment using 30 repeats. The values in the table represent the number of features that overlapped within the model under the given scenario (presented as mean (std)). The column values indicate the number of top features with overlap determined as the minimum threshold for significance. The maximum number of features for MFCC was 50. The absolute values of the feature importances were not zero. Abbreviations used: CHPS: characters per second, LRD: loud region duration.

Task	Scenario	TOP5 (min. 3)	TOP10 (min. 4)	TOP15 (min. 7)	TOP20 (min. 10)
Random Forest Classifier	pa vs. Italian	0.5 (±0.6)	1.5 (±0.8)	3.6 (±0.9)	6.7 (±1.5)
	pa vs. Eng	0.2 (±0.4)	0.8 (±0.7)	2.8 (±1.2)	5.1 (±1.1)
	Italian vs. Eng	0.0 (±0.1)	1.0 (±0.8)	3.2 (±1.0)	7.5 (±1.3)
Random Forest Regressor	Age vs. CHPS	1.2 (±0.4)	2.5 (±0.7)	5.4 (±1.0)	* 9.2 (±0.9)
	Age vs. LRD	1.0 (±0.0)	2.0 (±0.7)	4.6 (±1.0)	7.9 (±1.1)
	CHPS vs. LRD	2.1 (±0.3)	** 4.7 (±0.9)	** 8.7 (±0.8)	** 10.9 (±1.3)

* Indicates that significant overlap was achieved at least once by the threshold mean + 1σ. ** Indicates that significant overlap was achieved on average.

**Table 4 sensors-24-05520-t004:** Comparison of the selected literature-reported cross-db classification results obtained on dysarthria vs. HC datasets, relevant to the results obtained in this study.

Study	Features	Method	Acoustic Material (TRAIN)	Acoustic Material (TEST)	Metric ^1^	Drop ^2^
Hires et al., 2023 [50]	Traditional approach	XGBoost	CzechPD vowel /a/	RMIT-PD vowel /a/	AUC = 0.74	~7%
Hires et al., 2023 [50]	STFT	CNN	ITA ^3^ vowel /a/	RMIT-PD vowel /a/	AUC = 0.70	~25%
Ibarra et al., 2023 [73]	Mel-scale spectrograms	1D-CNN with DA ^4^	mixed four vowel /a/ datasets	PD-Neurovoz vowel /a/	Acc = 72%	0% ^5^
Ibarra et al., 2023 [73]	Mel-scale spectrograms	1D-CNN with DA ^4^	mixed four /pa-ta-ka/ datasets	PD-Neurovoz /pa-ta-ka/	Acc = 83%	3% ^5^
Tirronen et al., 2023 [74]	wav2vec 2.0	SVM	HUPA vowels /a/	SVD vowels /a/	Acc = 58%	~11%
Favaro et al., 2023 [45]	TRILLsson	PLDA + PCA	Cross-lingual SS/RP/TDU ^6^	CzechPD SS/RP/TDU ^6^	AUC was close to 1	0%
Javanmardi et al., 2024 [51]	Fine-tuned wav2vec 2.0-XLSR	SVM	EasyCall	TORGO	Acc = 70.3%	~1%
This paper	wav2vec 1.0	Random Forest	MDVR-KCL read text	ITA ^3^ read text	AUC = 0.90	8%
This paper	wav2vec 1.0	XGBoost	MDVR-KCL read text	ITA ^3^ read text	AUC = 0.98	0%
This paper	wav2vec 1.0	Random Forest	ITA ^3^ read text	MDVR-KCL read text	AUC = 0.72	8%
This paper	wav2vec 1.0	XGBoost	ITA ^3^ read text	MDVR-KCL read text	AUC = 0.78	2%

^1^ Achieved cross-db result under the specified setup. ^2^ When compared to intra-dataset classification. ^3^ Italian dataset [59], by Dimauro et al. ^4^ DA: domain adaptation; domain adversarial training. ^5^ Based on the presentation of results in the paper, we expect the target dataset was included as part of the mixed datasets used for training. ^6^ Spontaneous speech (SS) (e.g., monologue, picture description), reading passage (RP), and text-dependent utterances (TDUs).

## Data Availability

The Italian and English datasets analyzed during the current study are publicly available [59,60]. Italian data [59] are available at the following URL: https://ieee-dataport.org/open-access/italian-parkinsons-voice-and-speech (accessed on 22 July 2024). English data [60] are available at the following URL: https://zenodo.org/records/2867216 (accessed on 22 July 2024). The dataset comprised of participants rhythmically repeating the syllable /pa/ is not openly available at this moment and is available from the corresponding author upon reasonable request.

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
