# Peer review of "Analyzing Wav2Vec 1.0 Embeddings for Cross-Database Parkinson’s Disease Detection and Speech Features Extraction"

_sensors, 2024, doi:10.3390/s24175520_

Round 1
Reviewer 1 Report
Comments and Suggestions for Authors
The idea presented in this paper, "Analyzing Wav2Vec Embedding in Parkinson's Disease Speech: A Study on Cross-Database Classification and Regression Tasks” is good. The authors attempted to develop machine learning models for Parkinson’s Disease speech diagnosis tasks and analyzed feature importance on classification and regression tasks. The overall paper is well structured, however, the motivation and rationale behind performing both tasks are not clear. The authors are suggested to address the following comments while revising the paper.
General Comments:
1: It is unclear from the title, abstract introduction, and literature what specifically the authors are doing in this study. It is clear that regression and classification are being performed. But what particular variables are being targeted or what are the labels that authors are going to predict are not clear.
2: Keeping in view comment 1 revise the title to better reflect the work being done in this study. Also, discuss the key findings of the study and mention the best results achieved.
3: List down key research contributions being made in this study specifically highlighting the tasks.
4: Rewrite and organize the literature review based on classification and regression-based studies and also highlight the limitations associated with existing studies and research gaps that are being addressed in this study.
Technical Comments:
5: List down the details and descriptive statistics of the datasets for both tasks separately. Min, Max and mean of features other than textual features generated by speech conversion.
6: What is the number of examples used for classification and regression?
7: For the classification task give the number of examples in the dataset associated with each label.
8: Currently only Word2Vec is being used for feature embedding. What is the rationale behind using Word2Vec compared to GloVe, FastText, or BERT (Bidirectional Encoder Representations from Transformers), etc?
9: The authors are suggested to use the features using different embedding and compare the performance.
10: For classification only Random Forest is used? Why not logistic regression, SVM, or logistic regression? It is suggested to perform the experimentation using other models and also finetunes the hyperparameters.
11: Similarly for regression task, it is suggested to also use other models and then present the comparison.
12: What is the rationale for using 5-fold cross-validation is used in this study over 10-fold cross-validation or split validation?
13: How the importance of features is being compared for classification and regression where the tasks are different?
14: Compare and discuss the results in light of existing literature using the same dataset or the same features or techniques for the tasks being studied in this study.
15: Add AUROC graphs and confusion matrix for the classification results.
16: Figure 2, A, the x-axis shows that the dataset also contains age around 20, however in the introduction it is mentioned that the said disease is 65 and above. If age is being predicted based on the voice irrespective of the patients having disease, how is this task relevant to this study which is related to Parkinson's Disease speech?
Comments on the Quality of English LanguageMinor editing of English language required
Author Response
We would like to thank the reviewer for their valuable feedback and suggestions for improvement. We believe that we have significantly enhanced our manuscript and have addressed all the comments that required attention. Beyond responding to the reviewers' comments and suggestions, we have also made overall improvements to the manuscript. This includes expanding certain sections for better conceptual clarity, refining the language, and updating some of the terminology. All changes in the revised manuscript are highlighted in red.
General Comments:
1: It is unclear from the title, abstract introduction, and literature what specifically the authors are doing in this study. It is clear that regression and classification are being performed. But what particular variables are being targeted or what are the labels that authors are going to predict are not clear.
We have revised the title to better reflect the focus and findings of our study. Additionally, we have added highlights at the end of the Introduction section that outline the main contributions and the specific targets we aim to achieve. We hope these changes clarify the objectives and scope of our work.
2: Keeping in view comment 1 revise the title to better reflect the work being done in this study. Also, discuss the key findings of the study and mention the best results achieved.
We have included the best results in the abstract, as requested by Reviewer 2 as well.
3: List down key research contributions being made in this study specifically highlighting the tasks.
We have added highlights at the end of the Introduction section.
4: Rewrite and organize the literature review based on classification and regression-based studies and also highlight the limitations associated with existing studies and research gaps that are being addressed in this study.
We have added new literature and included current sources, but we haven't changed the structure of the review at this stage. The limitations and research gaps you mentioned are important and are addressed in the Main contributions of our study.
Technical Comments:
5: List down the details and descriptive statistics of the datasets for both tasks separately. Min, Max, and mean of features other than textual features generated by speech conversion.
A: After careful reiterating and considering this, we were not entirely sure how to address this comment. It's important to emphasize that our study does not involve working with textual features generated by speech conversion; please refer to our responses #8 and #9 for more context.
In general, we agree that it is very important to thoroughly characterize all datasets in their raw form and to present descriptive statistics. To address this, we have prepared Figure 2, which is placed in the first section of the Results. This figure illustrates the distributions of recording durations and loud region durations across the datasets used in the study. The mean, maximum, and minimum values are evident from this Figure.
6: What is the number of examples used for classification and regression?
A: Thank you for raising this question. We added more details in the Datasets section to include the number of subjects, explicitly stating labels and sample sizes for each task. For the regression task, we have also included the total number of subjects directly in the results section 4.2.
7: For the classification task give the number of examples in the dataset associated with each label.
A: Please, see our Answer #6.
8: Currently only Word2Vec is being used for feature embedding. What is the rationale behind using Word2Vec compared to GloVe, FastText, or BERT (Bidirectional Encoder Representations from Transformers), etc?
A: Thank you for this question.
We would like to clarify this. A distinction has to be made between speech-based wav2vec and other representations that operate on text, i.e. Natural Language Processing (NLP) (e.g., word2vec, GloVe, BERT, etc.), as these operate in different domains. Applying word2vec or similar NLP models to our experiments would first require running Automatic Speech Recognition, which is beyond the scope of this work.
Next, it is important to differentiate between speech-based feature extraction using wav2vec, which involves using extracted features directly, and Automatic Speech Recognition systems, such as wav2vec 2.0, which convert speech to text. The latter approach focuses on evaluating intelligibility based on the transcribed text. In our work, we use wav2vec features as a form of “tabular data” (i.e. fixed-length vectors) directly extracted from the speech signal, without processing text or performing transcription.
9: The authors are suggested to use the features using different embedding and compare the performance.
A: In terms of representations, we compare two approaches: self-supervised learning using wav2vec and a more traditional method of extracting audio features using MFCC. Additionally, we explore the fusion of these two representations and examine the effect of PCA on the combined features.
Regarding possible other speech-based embeddings, this work prioritizes wav2vec 1.0 over wav2vec 2.0. Despite its simpler design, wav2vec 1.0 has shown high performance, even compared to other transformer-based embeddings, as discussed in the paper.
10: For classification only Random Forest is used? Why not logistic regression, SVM, or logistic regression? It is suggested to perform the experimentation using other models and also finetunes the hyperparameters.
A: We have tested a new, higher-performing classifier. In this version, we present selected results (Figs. 3, 4, 5) using XGBoost. XGBoost was also employed as the primary classifier in a key study for this work to build upon, which we frequently cite in our work - Hires et al., 2023.
11: Similarly for regression task, it is suggested to also use other models and then present the comparison.
A:Thank you for pointing this out. Our primary intention with the regression analysis was not to evaluate improvements across different representations and classifiers, but rather to demonstrate that wav2vec is a viable method for modeling quantitative parameters in general. As part of this, we conducted a series of regression experiments using wav2vec that are relevant to PD, as shown in section 4.2. More importantly, we wanted to show that well-performing regression models are prerequisites for analyzing overlapping components.
In the current version, we have conducted a series of new experiments and present a comparison of overlapping components for wav2vec versus MFCC in both classification and regression, as seen in the new Tables 2 and 3.
12: What is the rationale for using 5-fold cross-validation is used in this study over 10-fold cross-validation or split validation?
A: Thank you for this question. Although 10-fold cross-validation is generally considered more robust than 5-fold cross-validation, the relatively small sample size for each dataset (~50 samples per dataset) led us to opt for 5-fold cross-validation. We prioritized 80-20 splits over 90-10 splits, which is closer to Leave-One-Out Cross-Validation. Additionally, in the revised version, we trained the models using full training and testing sets for selected outstanding cross-dataset results, as shown in Figures 3, 4, and 5.
13: How is the importance of features being compared for classification and regression where the tasks are different?
To analyze overlapping components between models, we performed statistical tests on the important features of both models. Specifically, we used the Fisher exact test to determine whether there is a significant overlap between the two set of features in terms of the number of overlapping features.
In Section 4.3, and more specifically in Figure 10, we implemented a straightforward approach to illustrate this. Please see the details below:
“We investigated the potential overlap between classification and regression tasks in the context of a binary Age classification model versus three regression tasks (Figure 10). The parameter Age was considered in two distinct forms: firstly, as a binary class (young vs. elderly), and secondly, as a quantitative regression model predicting age. Concurrently, two additional models, addressing entirely different aspects, were examined and tested against the important features derived from the Age classification model. The association between the number of features in the Age:classification vs. Age:regression model was found to be significant, indicating shared features between the tasks. In contrast, other cases were not found to be statistically significant and over-represented".
This strategy of analysis of overlapping components between classification and regression models may have limitations. However, the observed behavior aligns with our expectations, even in the absence of data-driven results.
14: Compare and discuss the results in light of existing literature using the same dataset or the same features or techniques for the tasks being studied in this study.
A: To facilitate comparison with the state of the art, we have designed and included Table 4 in the relevant part of Discussion. We also further extended the Discussion, reporting studies that recently worked with the same Italian and English datasets (inter-dataset).
“Several scenarios presented in Hires et al. [10.1016/j.ijmedinf.2023.105237] utilize the same Italian dataset as in this study, along with Czech Parkinsonian dataset (not the one used here), for both intra- and inter-dataset classification. The results are comparable and consistent with our findings, demonstrating very high accuracy for intra-dataset discrimination. Regarding other similar studies that have addressed classification using the same datasets (only inter-dataset), for the Italian dataset, we refer to the work by Malekroodi et. al [10.3390/bioengineering11030295]. The authors published a table comparing the accuracy results obtained from six approaches across five studies, all of which achieved an accuracy exceeding 95%. For the English dataset, we reference Cesare et al. [10.3390/s24051499], who compared multiple signal representations and classifiers, consistently obtaining accuracies close to or exceeding 90% in intra-dataset scenarios.”
15: Add AUROC graphs and confusion matrix for the classification results.
A: To address this comment, we performed a series of experiments, and the results are shown in Figures 3, 4, and 5.
16: Figure 2, A, the x-axis shows that the dataset also contains age around 20, however in the introduction it is mentioned that the said disease is 65 and above. If age is being predicted based on the voice irrespective of the patients having disease, how is this task relevant to this study which is related to Parkinson's Disease speech?
A: Thank you for pointing this out. In the Italian dataset, particularly in age modeling, we aimed to include all available data, including the young healthy control group, where the youngest participant is around 20 years old. We wanted to cover this age range in the regression model because it is highly relevant for analyzing early-onset Parkinson's disease.
Additional information can be found at [https://www.apdaparkinson.org/what-is-parkinsons/early-onset-parkinsons-disease, 10.3233/JPD-212815].
“When someone who is 21-50 years old receives a diagnosis of Parkinson’s disease, it is referred to as early onset Parkinson’s disease, or young onset Parkinson’s disease (YOPD).”
We have added information about early onset Parkinson’s disease to the Introduction.
Reviewer 2 Report
Comments and Suggestions for Authors
The Title: Analyzing Wav2Vec Embedding in Parkinson's Disease Speech: A Study on Cross-Database Classification and Regression Tasks.
This study employs the non-fine-tuned wav2vec 1.0 architecture to develop
machine learning models for PD speech diagnosis tasks, such as cross-database classification and regression to predict demographic and articulation characteristics. The primary aim is to analyze feature importance on both classification and regression tasks, investigating whether latent discrete speech representations in PD are shared across models, particularly for related tasks. The authors reflected their works faithfully and presented well organization. I believe it can be accepted after addressing the following issues.
Comments for the Authors:
1-It is preferable to remove the word “conclusion” from the Abstract.
2- What are the measurements that are used to assess the intelligibility that confirm the effectiveness of the proposed work? Please mention this point briefly in the Abstract.
Besides, intelligibility is considered one of the important attributes of speech signal in addition to the quality. Why there is no mention of them in the text body of the manuscript.
3- Is the work compared to other related works? Please mention this point briefly in the Abstract.
Furthermore, what is the percentage of accuracy improvement that has been obtained?
4- It is recommended to add a section for paper organization at the end of the introduction section.
5- There are many types of transformation in the literature that are used for feature extraction in addition to DTFT. They can be mentioned also in the related work section.
6- Some sentences are required between any section and its subsection for more clarity. Such as among section 3, subsection 3.1, and subsubsection 3.1.1
7- Any figures, data or equations should be cited with a reliable reference if they are taken from another source, such as the metrics in line 289.
8- What is the benefit of using the parameters “loud region duration” in regression tasks? Please give more explanation.
9- As mentioned in line, 458, “We constructed multiple models through various train-test combinations to evaluate the efficacy of wav2vec as a robust speech representation for PD detection.”. Comparison with recent existing works is essential to confirm the robustness of the proposed work in addition to the self-comparison.
Author Response
We would like to thank the reviewer for their valuable feedback and suggestions for improvement. We believe that we have significantly enhanced our manuscript and have addressed all the comments that required attention. Beyond responding to the reviewers' comments and suggestions, we have also made overall improvements to the manuscript. This includes expanding certain sections for better conceptual clarity, refining the language, and updating some of the terminology. All changes in the revised manuscript are highlighted in red.
1-It is preferable to remove the word “conclusion” from the Abstract.
A: The word “conclusion” has been removed.
2- What are the measurements that are used to assess the intelligibility that confirm the effectiveness of the proposed work? Please mention this point briefly in the Abstract.
A: Thank you for pointing this out. In the abstract, we intended to refer to articulation rather than intelligibility, although the two might be related. We have corrected this in the abstract.
Besides, intelligibility is considered one of the important attributes of speech signal in addition to the quality. Why there is no mention of them in the text body of the manuscript.
A: It is indeed true that intelligibility is considered one of the important attributes of the speech signal for evaluating speech in Parkinson's disease (as mentioned in the received comment). We recently studied this topic in another paper (https://ieeexplore.ieee.org/document/10605915). However, it is not directly relevant to this work. It focuses on Automatic Speech Recognition using wav2vec 2.0, which is directly associated with word recognition, and often validated using Word Error Rate.
3- Is the work compared to other related works? Please mention this point briefly in the Abstract.
Furthermore, what is the percentage of accuracy improvement that has been obtained?
A: Now, we explicitly mention that some cross-database wav2vec results demonstrated performance comparable to intra-dataset evaluations, which is a key finding of our study. Additionally, we have updated the abstract to reflect that we compared our results with other existing works (please see Table 4 in the new manuscript version).
4- It is recommended to add a section for paper organization at the end of the introduction section.
A: We added a section for paper organization at the end of the Introduction.
5- There are many types of transformation in the literature that are used for feature extraction in addition to DTFT. They can be mentioned also in the related work section.
In the "Related Work" section, in the paragraph starting with "DL methods have exhibited their efficacy in extracting valuable features from voice and speech...", we reference at least three of the most commonly used transformation methods that are subsequently fed into deep learning architectures. Specifically, these include STFT, Wavelet transformation, and Mel-spectrogram.
6- Some sentences are required between any section and its subsection for more clarity. Such as among section 3, subsection 3.1, and subsubsection 3.1.1
A: We have added some text between sections 3.1 and 3.1.1 for clarification. Regarding the suggestion to add text between sections 3 and 3.1, we believe the current structure is clear and self-explanatory, as 3.1. directly outlines the datasets used, and therefore, we suggest keeping it as is. To further enhance clarity of the structure for our obtained Results, we also added additional text in section 4. Results.
7- Any figures, data or equations should be cited with a reliable reference if they are taken from another source, such as the metrics in line 289.
A: Thank you for bringing this to our attention. We have thoroughly reviewed the manuscript and added references in several places in Methods, including the one mentioned above.
8- What is the benefit of using the parameters “loud region duration” in regression tasks? Please give more explanation.
A: Loud region duration is a parameter that can be approximated from raw signals and is directly associated with reading duration. Speech characteristics such as reading duration, pause determination, and speech instability are important factors in PD and are typically correlated with clinical scales as markers of motor impairment [10.1101/2024.02.23.24303241]. These parameters are generally calculated through manual inspection and annotation of sound files using the Praat software.
We also added this explanation to the relevant Methods section.
9- As mentioned in line, 458, “We constructed multiple models through various train-test combinations to evaluate the efficacy of wav2vec as a robust speech representation for PD detection.”. Comparison with recent existing works is essential to confirm the robustness of the proposed work in addition to the self-comparison.
A: We agree. To facilitate comparison with the state of the art, we have designed and included Table 4 in the relevant part of Discussion.
Round 2
Reviewer 1 Report
Comments and Suggestions for Authors
Thank you for making efforts to address the raised comments. Some points are not satisfactorily addressed in the revised version.
1. 4: Rewrite and organize the literature review based on classification and regression-based studies and also highlight the limitations associated with existing studies and research gaps that are being addressed in this study. 
 We have added new literature and included current sources, but we haven't changed the structure of the review at this stage. The limitations and research gaps you mentioned are important and are addressed in the Main contributions of our study. 
I am not able to find new literature being added or the section of related work being improved.
The authors are suggested to add a few studies from 2024 in the introduction and related work to cover the recent landscape of the domain such as:
Exploring the Impact of Fine-Tuning the Wav2vec2 Model in Database-Independent Detection of Dysarthric Speech." IEEE Journal of Biomedical and Health Informatics (2024).
Trends and Challenges in harnessing big data intelligence for health care transformation." Artificial Intelligence for Intelligent Systems: 220-240 (2024).
2. In Table 4: Favaro et. al, 2023 [44] the metric is stated as AUC >> 1, AUC value greater than 1 is not possible. Authors are suggested to carefully review the literature to report and refer to the findings.
3. Identify the limitations (very small data, serializability of findings?) and possible future direction of this study and add them at the end of the conclusion section.
Comments on the Quality of English LanguageMinor editing of English language required.
Author Response
Text edits in blue.
Thank you for making efforts to address the raised comments. Some points are not satisfactorily addressed in the revised version.
1. 4: Rewrite and organize the literature review based on classification and regression-based studies and also highlight the limitations associated with existing studies and research gaps that are being addressed in this study.
 We have added new literature and included current sources, but we haven't changed the structure of the review at this stage. The limitations and research gaps you mentioned are important and are addressed in the Main contributions of our study.
I am not able to find new literature being added or the section of related work being improved.
The authors are suggested to add a few studies from 2024 in the introduction and related work to cover the recent landscape of the domain such as:
Exploring the Impact of Fine-Tuning the Wav2vec2 Model in Database-Independent Detection of Dysarthric Speech." IEEE Journal of Biomedical and Health Informatics (2024).
Trends and Challenges in harnessing big data intelligence for health care transformation." Artificial Intelligence for Intelligent Systems: 220-240 (2024).
A: Thank you for suggesting the recent work "Exploring the Impact of Fine-Tuning the Wav2vec2 Model in Database-Independent Detection of Dysarthric Speech." Given its high relevance, we have included it in the Discussion: Table 4, even though it addresses dysarthria in general rather than specifically focusing on Parkinson’s disease. The following text (with one more study by the same research group being mentioned) was added to Related Work:
“In 2024, Javanmardi et al. advanced the field of inter-dataset dysarthric speech detection by exploring fine-tuned wav2vec 2.0 across three dysarthria datasets, with a layer-wise analysis [10.1109/JBHI.2024.3392829]. The same research group also conducted a systematic comparison between conventional features and pre-trained embeddings for dysarthria detection, demonstrating that intra-dataset pre-trained embeddings outperformed conventional features [10.1016/j.specom.2024.103047].”
Unfortunately, we are currently unable to access the second suggested publication, which appears to be a book. However, we have added another relevant 2024 study to the Introduction. We identified a similarly significant work titled Harnessing Voice Analysis and Machine Learning for Early Diagnosis of Parkinson's Disease: A Comparative Study Across Three Datasets. This study is particularly important due to its focus on early diagnosis, which is a major topic in the field today. Please see the text below:
“The integration of advanced machine learning techniques with voice analysis, validated across three diverse datasets, has shown significant potential for enhancing early detection of Parkinson's disease [10.1016/j.jvoice.2024.04.020].”
- In Table 4: Favaro et. al, 2023 [44] the metric is stated as AUC >> 1, AUC value greater than 1 is not possible. Authors are suggested to carefully review the literature to report and refer to the findings.

A: Thank you very much for spotting this. This was our mistake in the notation. Our intention was to indicate a value close to 1. In the revised version, we have corrected this, stating that the "AUC was close to 1", for this particular case.
- Identify the limitations (very small data, serializability of findings?) and possible future direction of this study and add them at the end of the conclusion section.

A: We added the following paragraph at the end of Conclusions.
“In our future research, we aim to address some of the limitations reported in this study. Specifically, we plan to validate our findings on more extensive datasets and further explore strategies for analyzing overlapping components between models and tasks. We will continue evaluating pretrained embeddings (such as wav2vec 1.0, wav2vec 2.0, and HuBERT) as universal representations that can accurately solve various problems in PD. Additionally, we intend to conduct a systematic comparison of different classification and regression techniques, followed by hyperparameter optimization.”